# Characteristics of Transient Flow in Rapidly Filled Closed Pipeline

**Kan Wang** [1,*] , **You Fu** [1] and **Jin Jiang** [2]

1 School of Energy and Power Engineering, Lanzhou University of Technology, Lanzhou 730050, China; fuyou_894513@hotmail.com
2 School of Power and Mechanical Engineering, Wuhan University, Wuhan 430072, China; jiangjing423@163.com
* Correspondence: 15193198471@163.com

**Abstract:** In this study, a one-dimensional mathematical model based on rigid theory is developed to evaluate the maximum water filling flow rate and filling time of closed pipeline water supply systems during rapid-filling processes. Polynomial fitting is utilized for prediction, and numerical simulation results are analyzed to understand the variations in maximum water filling flow rate, filling time, and pressure with respect to opening valve time, air valve area, filling head, and segmented filling pipe length. The findings highlight the significant impact of the filling head on the maximum water filling flow rate, while the filling time is predominantly influenced by the gas discharge coefficient. Rapid changes occur only at the initial stage of rapid filling, reaching the maximum value with a very high acceleration (around t = 4 s). It is observed that pressure fluctuations in the gas–liquid two-phase flow inside the pipeline lead to velocity differences and periodic changes in gas pressure opposite to the filling head. When the gas discharge coefficient reaches approximately 0.3, pressure variation within the water supply system diminishes, and the time and flow rate required for pipeline filling become independent of the discharge coefficient. This study suggests the use of a segmented filling approach to ensure the effectiveness and stability of pipeline filling.

**Keywords:** closed pipeline; filling time; maximum water filling flow rate; rapid filling; transient flow

## 1. Introduction

When a pipeline water transmission system is newly constructed or undergoing overhaul, its pipes initially contain no water. The first operational step involves opening gates, valves, and activating pumps along the pipeline to inject water. This procedure is essential for expelling air from the pipeline and ensuring it meets design requirements. Only after the pipeline is adequately purged of air can the system transition into its normal operational phase. This initial procedure is referred to as the water filling process [1]. The rapid filling of pipes represents a highly complex hydraulic transient phenomenon. During this process, the water flow undergoes several transitional phases, including open channel flow, gas–liquid two-phase flow, and a transition from non-pressurized to pressurized flow. This involves alternating between full pipe flow and a pressurized state. Therefore, the study of rapid water filling in pipeline systems is critically important to ensure the safe and energy-efficient operation of pipeline networks [2].

Recently, numerous experts and scholars have conducted research on the pipeline water filling process. Martin [3] used the water hammer equations to calculate the motion of a water column moving into a mass of entrapped air, illustrating the effect of the initial location of unconfined air pockets on the magnitude of the maximum air pressure, in which the water column is rigid. Cabrera et al. [4] found that the difference between the elastic model and the rigid model for a rapid-filling pipe is of less than 2%. Liou and Hunt [5] developed a model of a "rigid" water column to describe the unsteady motion of a lengthening water column filling an empty pipeline with an undulating elevation profile.

The water column velocity peaks early in the filling process. The peak velocity varies with the submergence at the pipeline inlet and the length of the initially static water column. Zhou et al. [6] conducted both experimental and analytical investigations into the pressure behaviors of stagnant air masses in fast-filling horizontal ducts, identifying three distinct patterns of pressure oscillation. Wang Ling [7] categorized the transient flow calculation models for pipeline filling into two main types: steady-state and dynamic models. Wang Ling further integrated the interface tracking method with a variable drag model for full pipe flow, developing a transient flow calculation model for pipeline filling that accounts for non-constant drag coefficients. Zhou Ling [8–10] and colleagues introduced an improved variable drag model into the simulation of water impacting stationary gas clusters. They developed a second-order Godunov finite volume model that incorporates variable drag. Utilizing the second-order Godunov scheme of the finite volume method, they established a discrete gas cavity model (DGCM) that considers dynamic drag. Their findings indicate that the interaction mechanisms within water–gas couplings are complex and can potentially lead to a maximum pressure value up to ten times the inlet pressure. Zhou Ling [11,12] et al. found that the initial gas volume and inlet pressure significantly influence the transient response in fast-filling horizontal pipelines. They proposed an analytical model capable of accurately capturing the pressure oscillations observed during the intermittent release of residual air. Furthermore, they developed an energy dissipation model for simulating the rapid filling of vertical pipelines containing trapped air.

A. Malekpour [13] developed a numerical model to simulate the dynamic aqueduct-filling process, starting from an initially empty state. This model effectively captures the transient changes that occur in the moving water column. Jiachun Liu [14] et al. provided a theoretical equation to describe the relationship between maximum air pressure (MAP) and the air integration coefficient. They also explored the influence of the air integration coefficient on MAP. Óscar E [15] formulated a mathematical model adept at simulating key hydraulic and thermodynamic variables in the filling process, particularly under the influence of an air valve in a single pipeline. Qingzhi Hou [16] et al. presented detailed experimental results on the characteristics of two-phase pressurized flow during the rapid filling of large pipelines. These results were then compared with those obtained from a typical one-dimensional rigid-column model. Ciro Apollonio [17] et al. tackled the issue of pressure transients that occur during the rapid filling of pipelines. They conducted experiments to mimic the filling process of pipelines under fluctuating conditions at atmospheric pressure. Duban A. Paternina-Verona [18] et al. analyzed the impact of varying initial conditions, specifically different oil and gas tank pressures and cavity sizes, on the rapid-filling operation of shaped PVC pipes. A. Tijsseling [19] focused on scenarios where stagnant gas masses have the ability to escape through vents, noting that, in such cases, the pressure in the liquid supply reservoir exhibits fluctuations instead of remaining constant.

In summary, the rapid filling of pipelines can lead to abnormal pressure fluctuations. Most existing research focuses primarily on improving mathematical models to analyze individual influencing factors in the rapid filling of closed pipelines. Predictions for unknown operating conditions are generally lacking, and there is limited research on transient flow during rapid filling under multi-variable conditions. This paper establishes a one-dimensional mathematical model based on the rigid-column theory to investigate the transient flow characteristics of rapid filling in closed pipelines. To evaluate the maximum filling velocity and filling time during the rapid-filling process in closed pipeline water delivery systems, a polynomial fitting method is used for prediction. Through orthogonal experimental analysis, polynomial fitting is employed to establish the relationship between the maximum filling flow rate, filling time, and influencing factors. The results indicate that the filling head has the most significant influence on the maximum filling flow rate, while the filling time is mainly affected by the gas discharge coefficient. This provides recommendations for effectively analyzing and controlling the transient flow during the filling process of pipeline systems.

## 2. Materials Methods

### 2.1. Rigid Model

Building on the foundational work of Liou and Hunt [5] and Axworthy and Karney [20], we developed and implemented a one-dimensional mathematical model. This model is based on four key assumptions: first, that the pipe cross-section remains fully filled during the filling process; second, that the pressure at the waterfront equates to atmospheric pressure; third, that the water–pipe system behaves as an incompressible entity (i.e., it forms a rigid water column); and fourth, that the friction within the system is quasi-steady. Let us denote Hi as the head at the downstream end of pipe $i$. Assuming that the waterfront advances in the $(i + 1)$th pipe, we can apply Newton's law of motion to the moving water column in the $(i + 1)$th pipe as follows [16]:

$$\frac{\partial Q}{\partial t} + Ag\frac{\partial H}{\partial x} + RQ|Q| = 0 \tag{1}$$

$$a^2\frac{\partial Q}{\partial x} + Ag\frac{\partial H}{\partial t} = 0 \tag{2}$$

$$\frac{\partial Q}{\partial t} + a^2\frac{\partial Q}{\partial x} + Ag\frac{\partial H}{\partial x} + RQ|Q| = 0 \tag{3}$$

$$\frac{DQ}{Dt} + Ag\frac{\partial H - H_{i+1}}{\Delta L} + RQ|Q| = 0 \tag{4}$$

$$\frac{l(t)}{gA_{i+1}}\frac{dQ}{dt} = H_i - \frac{f_{i+1}l(t)}{D_{i+1}}\frac{Q^2}{2gA_{i+1}^2} + l(t)\sin\theta_{i+1} \tag{5}$$

where $a$ = the wave speed; $R$ = resistance coefficient; $l(t)$ = length of the water column in the partially filled pipe; $\theta$ = angle downward from the horizontal; $f$ = Darcy–Weisbach friction factor; $A = \pi D^2/4$ is the pipe cross-sectional area; and $Q = VA$ is the uniform flow rate. Similarly, the fully filled upstream pipes yield the following:

$$\frac{L_j}{gA_j}\frac{dQ}{dt} = H_{j-1} - H_j - \frac{f_jL_j}{D_j}\frac{Q^2}{2gA_j^2} + L_j\sin\theta_j, j = 2,\ldots,i \tag{6}$$

where $i$ depends on $t$. Because of entrance head loss and velocity head at the inlet, the equation for the first pipe is slightly different, as follows:

$$\frac{L_1}{gA_1}\frac{dQ}{dt} = H_R - H_1 - (K + 1 + \frac{f_1L_1}{D_1})\frac{Q^2}{2gA_1^2} + L_1\sin\theta_1 \tag{7}$$

where $H_R$ = reservoir head; and $K$ = entrance loss coefficient. By adding Equations (4)–(6), all interior heads $H_j(j = 1,\ldots,i)$ are canceled, and one equation is obtained for the filling discharge $Q$:

$$\frac{1}{g}\left[\sum_{j=1}^{i}\frac{L_j}{A_j} + \frac{l(t)}{A_{i+1}}\right]\frac{dQ}{dt} = H_R - \left[\sum_{j=1}^{i}\frac{f_jL_j}{D_jA_j^2} + \frac{f_{i+1}l(t)}{D_{i+1}A_{i+1}^2} + \frac{K+1}{A_1^2}\right]\frac{Q^2}{2g} + \sum_{j=1}^{i}L_j\sin\theta_j + l(t)\sin\theta_{i+1} \tag{8}$$

Substituting the total length of the water column $L(t)$:

$$L(t) = \sum_{j=1}^{i}L_j + l(t) \tag{9}$$

$$\frac{1}{g}[C_1L(t) + C_2]\frac{dQ}{dt} = H_R - [C_3L(t) + C_4 + C_7]\frac{Q^2}{2g} + C_5L(t) + C_6 \tag{10}$$

where the time-dependent (because of $i$) coefficients $C_k(1,\ldots,7)$ are defined as follows:

$$C_1 = \frac{1}{A_{i+1}}, C_2 = \sum_{j=1}^{i} \frac{L_j}{A_j} - C_1 \sum_{j=1}^{i} L_j, C_3 = \frac{f_{i+1}}{D_{i+1} A_{i+1}^2}, C_4 = \sum_{j=1}^{i} \frac{f_j L_j}{D_j A_j^2} - C_3 \sum_{j=1}^{i} L_j, C_5 = \sin \theta_{i+1},$$

$$C_6 = \sum_{j=1}^{i} L_j \sin \theta_j - C_5 \sum_{j=1}^{i} L_j, C_7 = \frac{K+1}{A_1^2}$$

The total water column length is related to flow rate $Q$ by the following equation:

$$L = L_0 + \int_0^t \frac{Q}{A_{i+1}} dt \tag{11}$$

where $L_0(< L_1)$ = initial column length at $t = 0$ in Pipe 1. In Equations (9) and (10), $i$ is the index of the latest pipe that has been fully filled. It starts from 0, indicating that Pipe 1 is partially filled. When $L_0(< L_1)$ becomes larger than $L_1$, $i$ becomes 1. When $L(t)$ becomes larger than $L(t) > L_1 + L_2$, $i$ becomes 2, and so on.

Next, Equations (9) and (10) are solved for the two unknowns, $Q$ and $L$. Equation (9) is an ODE with the following initial condition:

$$Q(0) = 0 \tag{12}$$

Equation (10) is an integral equation in which $A_{i+1}$ depends on time. ODE Equation (9) is integrated with the fourth-order Runge–Kutta method and Equation (10) with Simpson's rule.

### 2.2. Polynomial Fitting Principle

The least squares method provides a solution to the challenge of seeking reliability from a set of measured values. Multiple sets of measurements with equivalent precision, encompassing valve-opening time, water head during filling, gas emission coefficient, pipeline length, maximum water filling velocity, and filling time data points are utilized with the aim of computing the optimal curve fitting the relationship between maximum water filling velocity and various factors, as well as a multifactor filling time curve.

Polynomial fitting is a method that utilizes polynomial functions to approximate real-world data. In the scenario involving four variables, consideration can be given to a quaternary polynomial function.

$$f(x_1, x_2, x_3, x_4) = \sum_{i=0}^{n} \sum_{j=0}^{n} \sum_{k=0}^{n} \sum_{l=0}^{n} a_{ijkl} \bullet x_1^i \bullet x_2^j \bullet x_3^k \bullet x_4^l \tag{13}$$

In this equation, $n$ represents the degree of the polynomial, $a_{ijkl}$ denotes the coefficients, and $x_1, x_2, x_3, x_4$ represents the four variables. To ensure that the fitted curve reflects the trend of the data, it is required that the sum of the squares of errors for all points is minimized, as follows:

$$\sum_{i=0}^{n} \sum_{j=0}^{n} \sum_{k=0}^{n} \sum_{l=0}^{n} (f(x_1, x_2, x_3, x_4) - V_{\max})^2 = \min \tag{14}$$

In this equation, $V_{\max}$ represents the observed data point for the maximum water filling velocity. To find its minimum value, the first derivative is equated to zero, as follows:

$$\frac{\partial f}{\partial x_1} = a_{10} + a_{11}x_2 + a_{12}x_3 + a_{13}x_4 = 0 \tag{15}$$

$$\frac{\partial f}{\partial x_2} = a_{01} + a_{11}x_1 = 0 \tag{16}$$

$$\frac{\partial f}{\partial x_3} = a_{02} + 2a_{12}x_1 + a_{23}x_4 = 0 \tag{17}$$

$$\frac{\partial f}{\partial x_4} = a_{03} + a_{13}x_1 + a_{23}x_3 + 2a_{33}x_4 = 0 \tag{18}$$

By solving the above equation for $a_{ijkl}$, the system of equations can be obtained.

### 2.3. Mathematical Model Validation

Experiments were conducted at the Fluid Machinery Laboratory of Lanzhou University of Technology to verify the accuracy of the mathematical model. The experimental system mainly consists of an upstream reservoir, a valve, an inclined pipeline, a horizontal pipeline, pressure sensors, and a downstream reservoir. The topographical features of the pipeline, including local high points and siphon pipes, were incorporated into the calculations. The total length of the pipeline is 200 m. The pressure sensors are located on the horizontal pipeline segment. It is assumed that the pipeline is initially empty, and the model does not account for the possibility of air entering the system. The simulation ends when the waterfront reaches the end of the pipeline. The initial pressure boundary condition is set at 40 m, with both the gas and liquid phases under atmospheric pressure. A comparison between the experimental and numerical calculation results is as follows (Figures 1 and 2):

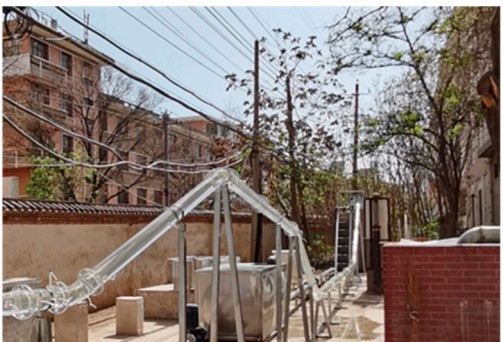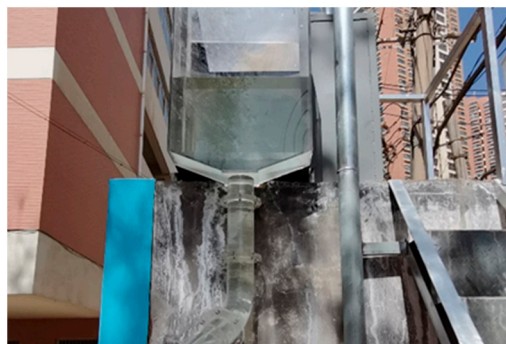

**Figure 1.** Schematic diagram of the experimental setup.

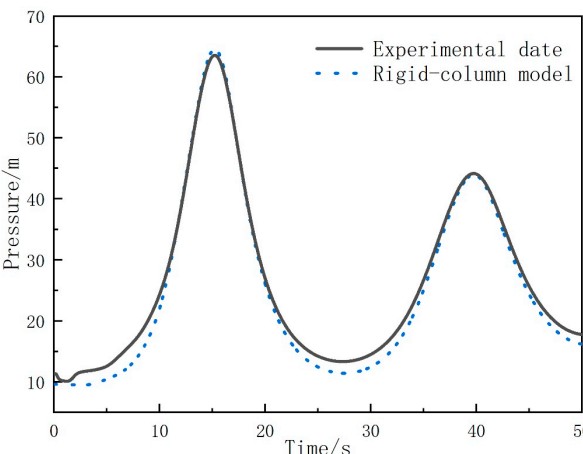

**Figure 2.** Experimental and numerical results.

The notable congruence between the rigid-column solution and the experimental results of fast-filling pipelines suggests that air intrusion plays a minimal role in the overall filling process [16]. Consequently, this numerical method proves to be a reliable tool for investigating the transient flow characteristics during the pipe-filling process.

### 3. Results and Discussion

#### 3.1. Multivariable Analysis of Closed Pipeline Filling

In this paper, a closed pipeline rapid-filling system as shown in Figure 3 is established to investigate the transient flow characteristics during rapid flushing of the pipeline. The pipeline system mainly consists of an upstream reservoir, valves, rigid closed pipelines, and downstream reservoirs. The inlet boundary pressure is set as the pressure inlet, the inner

diameter of the pipeline is 0.1 m, and the wave speed is 1000 m/s. During the simulation of the rapid-filling process, the monitoring and analysis of pressure, liquid column length, and velocity distribution within the pipeline are conducted by varying the boundary condition settings.

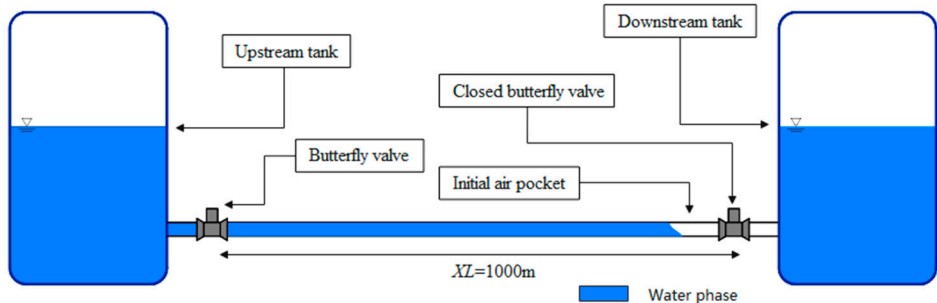

**Figure 3.** Schematic diagram of rapid filling of closed pipes.

In the rapid-filling process of a closed straight pipeline, the maximum water filling velocity under different conditions and the time required to fill the entire system are important indicators for determining the safety and economy of the filling method. Using orthogonal experimental design, calculations for the maximum water filling velocity and filling time are conducted for 25 sets of data. The results are presented in Table 1.

**Table 1.** Orthogonal experimental results.

| Factor | A Opening the Valve (s) | C Cd | P Boundary Pressure (m) | L Pipeline Length (m) | Maximum Velocity (m/s²) | Filling Time (s) |
|---|---|---|---|---|---|---|
| 1 | 0 | 0.01 | 10 | 100 | 9.05 | 227.3 |
| 2 | 0 | 0.2 | 30 | 300 | 16.00 | 28.33 |
| 3 | 0 | 0.5 | 50 | 500 | 20.73 | 32.43 |
| 4 | 0 | 0.7 | 70 | 700 | 24.56 | 40.47 |
| 5 | 0 | 1 | 100 | 1000 | 29.39 | 52.77 |
| 6 | 2 | 0.01 | 30 | 500 | 15.13 | 631.9 |
| 7 | 2 | 0.2 | 50 | 700 | 19.55 | 58.52 |
| 8 | 2 | 0.5 | 70 | 1000 | 23.08 | 66.31 |
| 9 | 2 | 0.7 | 100 | 500 | 27.40 | 23.47 |
| 10 | 2 | 1 | 10 | 700 | 8.74 | 113.34 |
| 11 | 4 | 0.01 | 50 | 1000 | 18.91 | 905.8 |
| 12 | 4 | 0.2 | 70 | 100 | 21.07 | 7.43 |
| 13 | 4 | 0.5 | 100 | 300 | 26.18 | 14.87 |
| 14 | 4 | 0.7 | 10 | 500 | 8.50 | 76.67 |
| 15 | 4 | 1 | 30 | 700 | 14.79 | 64.81 |
| 16 | 6 | 0.01 | 70 | 300 | 21.11 | 213 |
| 17 | 6 | 0.2 | 100 | 500 | 25.36 | 28.75 |
| 18 | 6 | 0.5 | 10 | 700 | 8.33 | 118.8 |
| 19 | 6 | 0.7 | 30 | 1000 | 14.44 | 103.6 |
| 20 | 6 | 1 | 50 | 100 | 18.29 | 8.8 |
| 21 | 8 | 0.01 | 100 | 700 | 24.67 | 383.7 |
| 22 | 8 | 0.2 | 10 | 1000 | 8.17 | 213.2 |
| 23 | 8 | 0.5 | 30 | 100 | 13.55 | 11.75 |
| 24 | 8 | 0.7 | 50 | 300 | 17.86 | 22.27 |
| 25 | 8 | 1 | 70 | 500 | 21.04 | 31.28 |

To assess the extent of the influence of factors on the performance indicators, a range analysis is conducted on the experimental results, as shown in Table 2. Here, $K_i$ ($i$ = 1, 2, 3) represents the average value of all experimental results corresponding to the ith level

of a certain indicator factor, and R denotes the range, which is the difference between the maximum and minimum values of $K_i$. A larger difference indicates a greater influence of the factor on the evaluation indicator.

**Table 2.** Range analysis.

| $K_i$ | Maximum Velocity (m/s$^2$) | | | | Filling Time (s) | | | |
|---|---|---|---|---|---|---|---|---|
| | **A** | **C** | **P** | **L** | **A** | **C** | **P** | **L** |
| $K_1$ | 19.9 | 17.8 | 8.6 | 17.9 | 76.3 | 472.3 | 149.9 | 55.8 |
| $K_2$ | 18.8 | 18.0 | 14.8 | 18.0 | 178.7 | 67.2 | 168.1 | 78.4 |
| $K_3$ | 17.9 | 18.4 | 19.1 | 18.2 | 213.9 | 48.8 | 205.6 | 160.2 |
| $K_4$ | 17.5 | 18.6 | 22.2 | 18.4 | 94.6 | 53.3 | 71.7 | 133.3 |
| $K_5$ | 17.1 | 18.4 | 26.6 | 18.8 | 132.4 | 54.2 | 100.7 | 268.3 |
| R | 2.9 | 0.8 | 18.0 | 0.5 | 137.7 | 423.5 | 133.9 | 212.6 |
| Factor priority | P > A > C > L | | | | C > L > A > P | | | |
| Optimal group | $P_5A_1C_4L_5$ | | | | $C_3L_1A_1P_4$ | | | |

The calculation method for $K_i$ is as follows:

$$K_i = \sum_{i=1}^{n} X_i; k_i = \frac{\sum_{i=1}^{n} X_i}{n} \tag{19}$$

where $n$ is the number of experimental results at the $i$-th level. $X_i$ represents each individual experimental result at the $i$-th level.

Example: For the factor "valve-opening time" at level 1, when the valve-opening time is 0 min:

$$K_1 = 9.05 + 16.00 + 20.73 + 24.56 + 29.39 = 99.73; k_1 = \frac{1}{5}K_1 = 19.9 \tag{20}$$

According to Table 2, it is evident that the factors influencing the maximum water filling velocity are ranked as follows: water head during filling > valve-opening time > gas discharge coefficient > pipeline length. This implies that a greater water head during filling, shorter valve-opening time, larger gas discharge coefficient, and longer pipeline length result in a higher maximum water filling velocity. Regarding the influence on filling time, the order of factors is as follows: gas discharge coefficient > pipeline length > valve-opening time > water head during filling. From the comprehensive analysis of these results, it can be observed that a shorter valve-opening time (Factor A) is beneficial for both performance indicators.

The method of variance analysis using the F-test, which employs the F distribution to determine significance, is employed. Both the F distribution and the normal distribution are continuous probability distribution models, but the F distribution is asymmetric and has two parameters: the degrees of freedom for factors and the degrees of freedom for errors. Typically, the assumption is initially made that the influence of factors is not significant, and then the ratio of factor variance to error variance is compared. By conducting variance analysis using the F-test on the experimental results, we can analyze the extent to which control conditions and random errors affect the results, thereby determining the significance of controllable conditions on the research results. This allows for a better evaluation of the significance of each factor's influence on performance indicators. The results of variance analysis for the optimal combination are shown in Table 3.

**Table 3.** Analysis of variance.

| Performance Indicator | Variance Source | Sum of Squares of Deviations | Degrees of Freedom | F Ratio | Critical F Value |
|---|---|---|---|---|---|
| Maximum velocity | A | 26.260 | 4 | 0.106 | 2.33 |
| | C | 2.095 | 4 | 0.008 | 2.33 |
| | P | 958.534 | 4 | 3.874 | 2.33 |
| | L | 2.717 | 4 | 0.011 | 2.33 |
| | Error | 989.610 | 16 | - | - |
| Filling time | A | 65,702.773 | 4 | 0.275 | 2.33 |
| | C | 694,652.544 | 4 | 2.905 | 2.33 |
| | P | 56,948.211 | 4 | 0.238 | 2.33 |
| | L | 139,089.027 | 4 | 0.582 | 2.33 |
| | Error | 956,392.560 | 16 | - | - |

From Table 3, it can be observed that the significance influence sequence derived from variance analysis is generally consistent with the sequence obtained from range analysis. Through variance analysis, we can further clarify the significance of factors on each indicator. The water head during filling has a significant impact on the maximum water filling velocity but has an insignificant effect on the filling time. The gas discharge coefficient significantly affects the filling time.

*3.2. Analysis of the Impact of Valve-Opening Time*

To validate the accuracy of the significance analysis in the orthogonal experiment, let us assume that the water head in the pipeline system is 50 m, the gas discharge coefficient Cd is 0.05, and the butterfly valve-opening time is adjusted to 0.0 s (instantaneous opening), 2.0 s, 4.0 s, 6.0 s, and 8.0 s, respectively. In this way, during the water filling process in the pipeline system, the water filling velocity at each moment increases as the valve-opening time decreases. The variation in water filling velocity within the pipeline at different valve-opening times is shown in Figures 3 and 4.

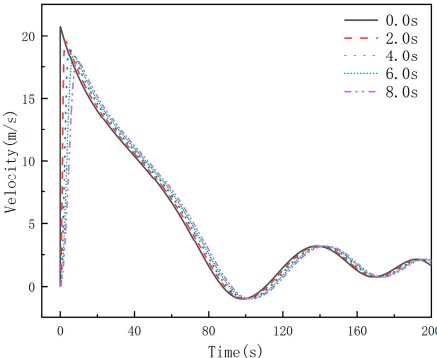

**Figure 4.** Variation in water filling velocity at different valve-opening times.

Figures 4 and 5 depict the variation in water filling velocity over time as the butterfly valve-opening time changes from 0.0 s to 8.0 s. According to the computational results, during rapid filling, the liquid column moves through the system as a rigid column of water. Rapid changes occur only at the initial stage of filling, reaching maximum velocity with very rapid acceleration. Subsequently, due to the increase in the length of the liquid column (resulting in increased inertia and surface friction), the velocity and flow gradually decrease. The contributions of driving force and resistance vary at different stages of filling. As the valve-opening time changes from 0.0 s to 8.0 s, the water filling velocity increases with the shortening of the valve-opening time at each moment. Over time, the differences in water filling velocity corresponding to different valve-opening times decrease. Since the relative change in water filling velocity is small, it has a minor impact on the pipeline filling time, but the pressure within the water distribution system remains unchanged. To

quantify the differences in computational results, statistics on the maximum water filling velocity and filling time for different valve-opening times are presented in Table 4.

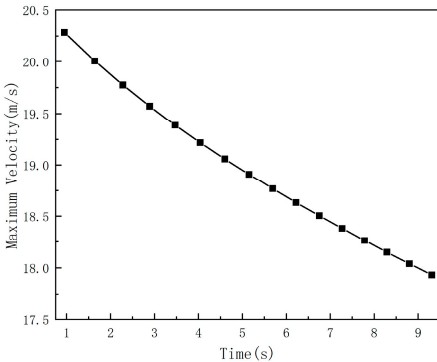

**Figure 5.** Valve-opening time—maximum water filling velocity curve.

**Table 4.** Statistics of maximum water filling velocity and filling time under different valve-opening times.

| Opening the Valve | Time | Maximum Velocity (m/s) | Time/Opening the Valve | Filling Time |
|---|---|---|---|---|
| 0 | 0.06 | 20.730 | 0.00 | 206.9 |
| 4 | 5.15 | 18.909 | 0.78 | 209.3 |
| 6 | 7.27 | 18.382 | 0.83 | 210.5 |
| 8 | 9.31 | 17.932 | 0.86 | 211.7 |

From Table 4, it can be observed that the discrepancy in calculating the maximum water filling velocity is greater than that in calculating the filling time. Moreover, for both the maximum water filling velocity and filling time, the maximum discrepancies occur at 0.0 s and 2.0 s, respectively. The maximum discrepancy in water filling velocity is 2.83 m/s, while for filling time, it is 1.3 s.

### 3.3. Analysis of the Influence of Water Head on Performance

To further investigate the influence of rigid models under different conditions on the numerical results of transient water filling, an analysis was conducted on the impact of varying water heads on the parameters of rapid-filling flow in closed pipelines when the butterfly valve was opened for 0.5 s. For the closed pipeline, calculations were performed based on different initial pressure boundaries to obtain the corresponding relationship between flow parameters inside the pipeline and time. The following figure illustrates the pressure characteristics and velocity distribution of the closed pipeline rapidly filling to full status under different water heads. See Figures 6 and 7.

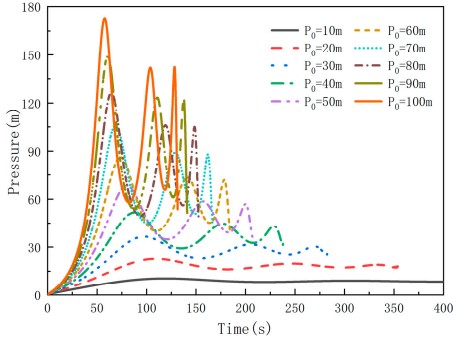

**Figure 6.** Pressure variation under different water head conditions.

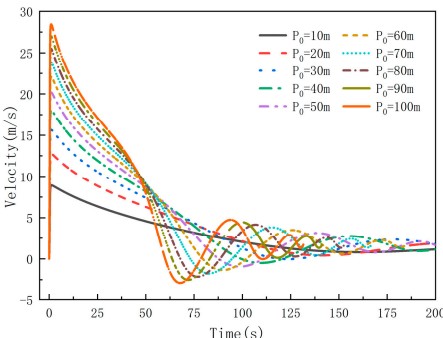

**Figure 7.** Variation in water inflow velocity under different water head conditions.

From the above figure, it can be seen that under different water head conditions, the trends of various flow parameters over time are basically consistent. The interaction between the moving liquid column and the valve leads to the generation of peak pressure. With an increase in the initial boundary pressure value, the time required for the pipeline to fill decreases. However, for the same initial boundary pressure change, the relative change in the overall filling time is smaller. The time taken for the pressure to reach its maximum value is shorter, and the rate of pressure increase is faster (except for $P_0 = 10$ m). Rapid changes only occur during the initial rapid-filling stage, reaching maximum values with very fast acceleration (around $t = 4$ s). Subsequently, the magnitude of the water inflow velocity corresponding to different operating conditions decreases as the water head decreases, while the period of water inflow velocity increases. To further analyze the mechanism of periodic changes in water inflow velocity, this paper conducts computational analysis based on the changes in flow parameters when the water head is 100 m, as shown in Figure 8.

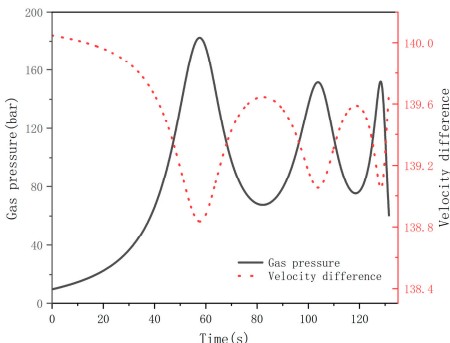

**Figure 8.** Relationship between gas pressure and velocity difference.

From Figure 8, it can be observed that during the filling of a closed straight pipe, the distribution of velocity difference ($\Delta H = \frac{a\Delta v}{g}$) within the pipe is exactly opposite and has the same periodicity as the change in gas pressure. This phenomenon is caused by the pressure fluctuations in the gas–liquid two-phase flow inside the pipe. As the pipe starts to fill with water, the liquid gradually displaces the gas inside the pipe. During this process, the flow of liquid causes changes in the gas pressure inside the pipe. As liquid continues to enter the pipe, the volume of gas decreases, leading to an increase in gas pressure inside the pipe. Meanwhile, due to the different flow patterns of gas–liquid two-phase flow inside the pipe, the distribution of liquid flow velocity is opposite in periodic fluctuations to the change in gas pressure. When gas pressure increases, the liquid velocity may decrease because the increase in gas pressure hinders the flow of liquid. Conversely, when gas pressure decreases, the liquid velocity may increase because the decrease in gas pressure promotes the flow of liquid. The phenomenon where the distribution of velocity difference between gas and liquid phases within the pipe is exactly opposite and has the same periodicity as the change in gas pressure is caused by the dynamic interaction

between liquid flow and gas pressure fluctuations. From Figure 9, it can be seen that as the initial boundary pressure value increases, the time required to fill the pipe decreases, but for the same initial boundary pressure change, the relative change in the overall filling time decreases. When the boundary pressure is higher, the maximum velocity of the pipe system also increases, reducing the time required to fill the pipe. The increase in boundary pressure has a decreasing effect on the maximum velocity and filling time of rapid pipe filling. The fitting equation is as follows:

$$V_{\max} = 4.861 + 0.453P - 0.358 \times 10^{-2}P^2 + 1.428 \times 10^{-5}P^3 \tag{21}$$

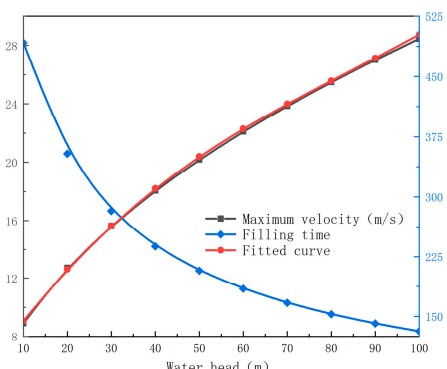

**Figure 9.** Variation in flow characteristics under different water head conditions.

To quantify the effect of water head on the rapid filling of the pipeline, statistical analysis was conducted on the peak pressure and maximum water head for different water head conditions, as shown in Table 5.

**Table 5.** Statistical analysis of peak pressure for different water head conditions.

| Boundary Pressure (m) | Peak Pressure (m) | $P_{\max}/P_0$ | Relative Change | Maximum Velocity |
|---|---|---|---|---|
| 10 | 10.52 | 1.05 | | 8.91 |
| 20 | 22.78 | 1.14 | 0.087 | 12.73 |
| 30 | 36.52 | 1.22 | 0.078 | 15.63 |
| 40 | 51.70 | 1.29 | 0.075 | 18.06 |
| 50 | 68.30 | 1.37 | 0.074 | 20.18 |
| 60 | 86.33 | 1.44 | 0.073 | 22.10 |
| 70 | 105.77 | 1.51 | 0.072 | 23.85 |
| 80 | 126.65 | 1.58 | 0.072 | 25.49 |
| 90 | 148.96 | 1.66 | 0.072 | 27.02 |
| 100 | 172.76 | 1.73 | 0.072 | 28.46 |

Peak pressure and maximum velocity under different water head conditions are statistically analyzed, as shown in Table 5. Both maximum pressure and velocity increase with increasing water head. It can be observed that when the water head is $P_0 = 10$ m, the minimum value of $P_{\max}/P_0$ is 1.05 and the minimum velocity is 8.91 m/s; when the water head is $P_0 = 100$ m, the maximum value of $P_{\max}/P_0$ is 1.73, and the maximum velocity is 28.46 m/s. The relative change in peak pressure when the initial boundary pressure increases from 10 m to 20 m is approximately 0.087 at maximum, and approximately 0.072 at minimum. The maximum change in velocity when the initial boundary pressure increases from 10 m to 20 m is approximately 3.83, and when it increases from 90 m to 100 m, it is approximately 1.44. Therefore, the change in velocity is negatively correlated with the water head, and the increase in water head gradually decreases the maximum velocity of rapid pipeline filling, with the influence on peak pressure being minimal.

### 3.4. Analysis of the Impact of Cd

The gas emission coefficient is a constant representing the flow capacity of the valve. In order to facilitate the analysis of the impact of the gas emission coefficient on the rapid filling of the pipeline, and to significantly demonstrate the changing trends of flow characteristics with different Cd values, 12 parameters ranging from 0.01 to 1.0 are selected. Under the conditions of constant water head = 50 m, pipe length L = 1000 m, and pipe diameter = 1000 mm, calculations are performed for the pipeline pressure and water inflow velocity during the filling process until the pipeline is fully filled.

The curves depicting the pressure and velocity variation over time under different discharge coefficient conditions are shown in Figures 10 and 11, respectively. From the graphs, it can be observed that as the discharge coefficient decreases, indicating a reduction in the flow capacity of the air valve and a decrease in gas discharge, the pressure inside the pipe shows an increasing trend. Consequently, more time is required to fill the pipe, while the maximum pressure remains relatively consistent, approximately 0.03–1.37 times the upstream water head. During the initial phase of water filling, when the liquid column experiences high acceleration, rapid filling can yield significant effects. At this stage, the gas has not yet been dominantly expelled through the air valve, resulting in consistent velocity changes within the first 0 to 5 s, with a maximum water inflow velocity of approximately 20.18 m/s, reaching peak flow rate. Subsequently, the velocity variation is positively correlated with the discharge coefficient after the air valve comes into play.

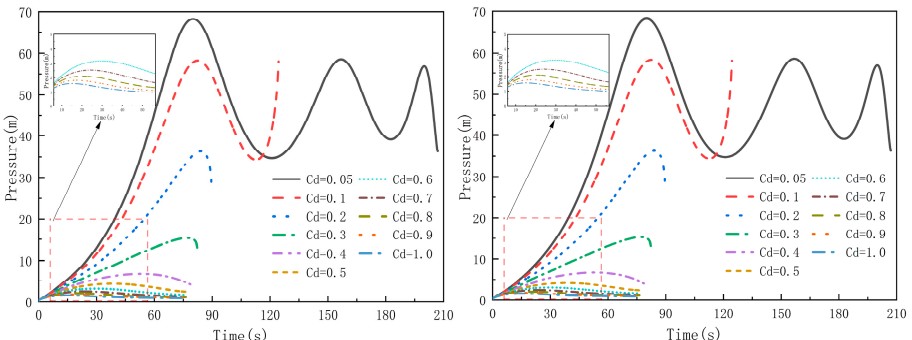

**Figure 10.** The pressure variation under different discharge coefficients.

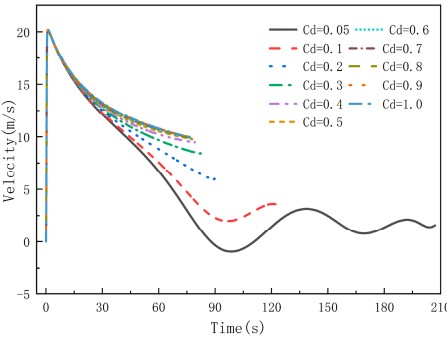

**Figure 11.** The velocity under different discharge coefficients.

As the water column lengthens, the acceleration decreases, and hydraulic losses control the velocity of the filling front. Figure 11 is presented to further analyze the relationship between maximum water inflow velocity, filling time, and discharge coefficient.

From Figure 12, it can be observed that the effect of the gas discharge coefficient on the maximum velocity is minimal. When Cd varies from 0.05 to 1, the moment at which the maximum velocity occurs remains unchanged at 1.1 s, with an increase in value of 0.01 m/s. As the discharge coefficient decreases, indicating a reduction in the flow capacity of the air valve, more time is required to fill the pipeline. When Cd is approximately ≥ 0.3,

the pressure variation within the water delivery system is small, and the filling time and water inflow velocity are no longer affected by the discharge coefficient.

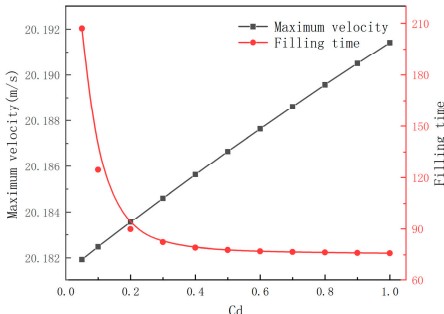

**Figure 12.** The variation in flow characteristics under different discharge Cd.

The fitting formula for the filling time and the gas discharge coefficient is as follows:

$$t = 219.008 - 820.337C + 1404.517C^2 - 735.074C^3 \tag{22}$$

### 3.5. The Influence of Sectional Water Filling on Rapid Pipeline Filling

In practical engineering applications, there may be situations where the pipeline is not completely filled. In such cases, sectional water filling can be adopted to fill the pipeline. Therefore, under the conditions of a constant water head of 50 m, a discharge coefficient (Cd) of 0.03, and a pipe diameter of 1000 mm, ten different groups of pipe lengths are selected for analysis and calculation. The relationship between pressure and velocity distribution over time for different pipe lengths is obtained, as shown in Figure 12.

From Figure 13, it can be observed that under different pipeline lengths, the trends of pressure and velocity variation over time are generally consistent: pressure alternates between peaks and valleys, with a relatively slower increase in pressure for longer pipelines. During the initial phase of water filling, when the liquid column experiences high acceleration, rapid filling can yield significant effects. Only the initial boundary pressure value affects the maximum velocity; so, the maximum velocity remains the same during sectional water filling. After reaching maximum velocity, longer pipelines experience lower acceleration and thus longer filling times. Considering both the filling time and peak pressure, the influence of sectional water filling on rapid filling of closed pipelines cannot be ignored.

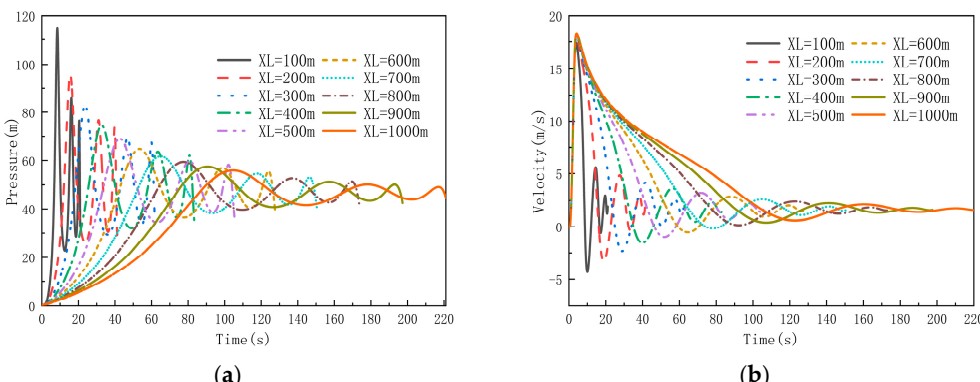

**Figure 13.** Variability in flow characteristics of segmental water filling. (**a**) Pressure characteristics; (**b**) Velocity distribution.

The influence of various factors on the flow characteristics during the rapid-filling process of a closed pipeline is analyzed from different perspectives to further validate the accuracy of the significance analysis conducted through orthogonal experiments. The initial value of boundary pressure significantly affects the maximum filling flow rate, while the filling time is most influenced by the gas discharge coefficient.

## 4. Conclusions

This study is based on the rigid-column theory to establish a one-dimensional mathematical model. Through orthogonal experiments, the significant factors influencing the flow characteristics during the rapid-filling process of closed pipeline water delivery systems are analyzed. Polynomial fitting is utilized to establish the relationships between the maximum filling velocity, filling time, and the influencing factors, providing predictive evaluations. The following conclusions are drawn:

(1) During the rapid-filling process, the liquid column, acting as a rigid column, moves within the system. Rapid changes only occur in the initial phase of filling, reaching maximum values with very high acceleration. As the length of the liquid column increases (thus increasing inertia and surface friction), the velocity and flow rate gradually decrease. Among these, the initial boundary pressure significantly affects the maximum water inflow velocity, while the filling time is most influenced by the gas discharge coefficient.

(2) When the valve-opening time varies, the maximum water inflow velocity changes accordingly, affecting the pipeline filling time. However, the relative change in water inflow velocity is small, and there is no change in the pressure values within the water delivery system. This is because the pressure fluctuations in the gas–liquid two-phase flow inside the pipeline cause the distribution of velocity difference to be exactly opposite in periodic fluctuations to the change in gas pressure.

(3) When Cd is approximately $\geq 0.3$, the pressure variation within the water delivery system is small, and the filling time and water inflow velocity are no longer affected by the discharge coefficient. In practical engineering applications, sectional water filling can be adopted to fill the pipeline in cases where the pipeline is not completely filled.

We did not fully consider the operational speed limitations in our study and did not investigate the specific conditions of segmented filling. In subsequent research, we will more thoroughly incorporate these operational constraints to ensure that this study is more accurate and has practical relevance. A preliminary analysis indicated that the impact of unsteady friction on the outcomes was minimal, and that further research could improve the model.

**Author Contributions:** Methodology, Y.F.; writing—original draft preparation, K.W.; writing—review and editing, J.J. Funding acquisition, Y.F. All authors have read and agreed to the published version of the manuscript.

**Funding:** This research was funded by the National Science Foundation for Distinguished Young Scholars of China (52009051) and the Gansu Province Natural Science Foundation of China (23JRRA800).

**Data Availability Statement:** Data will be made available on request.

**Conflicts of Interest:** There are no conflicts of interest.

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
