# Peer review of "Characteristics of Transient Flow in Rapidly Filled Closed Pipeline"

_water, doi:10.3390/w16172377_

Round 1

Reviewer 1 Report (New Reviewer)

Comments and Suggestions for Authors

The study presents a comprehensive analysis that details the various factors impacting the efficiency of rapid filling processes. These factors include the timing of valve openings, the size of the air valve area, the design of the filling head, and the length of segmented filling pipes. Additionally, the research provides practical recommendations aimed at improving the reliability of water supply systems during these rapid filling scenarios. A notable aspect of this study is its methodological rigor, emphasized by the employment of polynomial fitting and orthogonal experimental analysis to ensure the robustness of the findings. In order to improve further below suggestions and recommendations needs to be incorporated.

·       The author should include the quantitative major outcomes in the abstract.

·       The study does not fully consider operational speed limitations and specific conditions of segmented filling, which could affect the practical applicability of the results.

  • The research focuses primarily on mathematical modelling and lacks experimental validation under real-world conditions.

Author Response

Reviewer 2 Report (New Reviewer)

Comments and Suggestions for Authors

The observation results and the analysis conducted concern an interesting phenomenon. The article employs a reasonable approach with clear and logical reasoning, making it easy to read and understand. The content holds a certain reference value. However, there are a few areas that need to be addressed in the paper:

- The article uses an orthogonal experiment to perform a significance analysis on multiple factors involved in the rapid filling of a closed pipeline. Could you explain why a single-factor influence analysis on factors such as valve opening time is conducted even after performing the significance analysis using orthogonal analysis?

- In the text, the rigid theory is used for numerical calculations. Please explain what the advantages of this theory are.

- How was this equation developed? Also, what is this C variable? Is this Cd?

These points need clarification to enhance the understanding and robustness of the paper.

Author Response

Reviewer 3 Report (New Reviewer)

Comments and Suggestions for Authors

The authors presented experimental and theoretical investigations on the dynamics of water filling in a pipe. The paper studied the effects of different variables (opening valve, Cd, boundary pressure, pipe length) on the maximum velocity and filling time.

I find the application of a one dimensional model to the filling of a closed pipe to be questionable. As the gas pressure increases, the first and second assumptions of the model are not compatible. The increase gas pressure will break down the one-dimensional assumption of the model. More data needs to be shown to prove this assumption is sound.

Writing issues:

When equations are laid out specifically, the meaning of each equation from 1 to 5 should be expounded in the text. The symbols are not explained fully. What is ‘a’, ‘R’ in equation 2? Why is absolute value used on Q?

The statistical methods need to be better explained. For example, how are Ki calculated?

Round 2

Reviewer 3 Report (New Reviewer)

Comments and Suggestions for Authors

Ready for publication

Author Response

Dear Professor:

   In this study, a one-dimensional mathematical model based on rigid theory is developed to evaluate the maximum water filling flow rate and filling time of closed pipeline water supply systems during rapid filling processes. Polynomial fitting is utilized for prediction, and numerical simulation results are analyzed to understand the variations in maximum water filling flow rate, filling time, and pressure with respect to opening valve time, air valve area, filling head, and segmented filling pipe length.

  Thank you for dedicating your time to review my paper and provide valuable feedback. I highly appreciate your input, and I have carefully considered your comments while revising my work.

  Best regards,

  Kan Wang.

This manuscript is a resubmission of an earlier submission. The following is a list of the peer review reports and author responses from that submission.

Round 1

Reviewer 1 Report

Comments and Suggestions for Authors

The author evaluated the maximum filling velocity and filling time of closed pipeline water conveyance systems during the rapid filling process, using polynomial fitting methods for prediction. The article employs a reasonable approach with clear and logical reasoning, making it easy to read and understand. The content holds a certain reference value. However, there are a few areas that need to be addressed in the paper:

- The article uses an orthogonal experiment to perform a significance analysis on multiple factors involved in the rapid filling of a closed pipeline. Could you explain why a single-factor influence analysis on factors such as valve opening time is conducted even after performing the significance analysis using orthogonal analysis?

- The rigid theory is employed for numerical calculations in the article. Please explain the feasibility of this theory.

- In the text, "air" and "gas" should be consistently referred to using the same term.

- There is a formatting issue with the references "[19][20]" on page 63, lines 2 and 3.

Comments on the Quality of English Language

it can be improved.

Reviewer 2 Report

Comments and Suggestions for Authors

The observation results and the analysis conducted concern an interesting phenomenon. The only doubts are raised by the description of the two-phase flow phenomenon (line 269). The description is quite laconic, and the analysis of two-phase motion parameters is too simplified. However, due to the length of the article, it was probably impossible to conduct a deeper analysis.

Reviewer 3 Report

Comments and Suggestions for Authors

Please refer to the attached PDF file

Comments on the Quality of English Language

English is unclear and clunky at points. There are many points in the manuscript without proper use of units, an explanation of why there are so many significant digits and a strong need to structure results in a way that makes some sense. 

Round 2

Reviewer 3 Report

Comments and Suggestions for Authors

While minor corrections were added (e.g., adding units to tables), many corrections were not. Some examples:

1) My comment on the experimental data collection: vaguely responded as "Using this experimental data to validate the applicability of the numerical method presented in this paper." Which "this data" are you referring to? Which source/reference for this data?

2) If the downstream tank is not represented in the formulation, why is it represented in Figure 2? The downstream tank there is filled, and there is indeed an indication of an air pocket in the figure, unlike what the authors stated.

3) The authors neglected the request for re-structuring the manuscript completely so that it follows the normal sequence "Introduction/Objectives - Methods - Results and Discussion - Conclusions and Recommendations". As is, this remains a very confusing work. Particularly, it is unclear how the sections are articulated to support the establishment of "... a one-dimensional mathematical model based on the rigid column theory to investigate the transient flow characteristics of rapid filling in closed pipelines." Also, what contents of the sections 4 and 5 are methodology, and what constitute research results.

Comments on the Quality of English Language

There has been no indication that the English was edited from the previous manuscript.
